# Associations between Nutrigenomic Effects and Incidences of Microbial Resistance against Novel Antibiotics

**DOI:** 10.3390/ph16081093

**Published:** 2023-08-01

**Authors:** Mohamed A. Raslan, Sara A. Raslan, Eslam M. Shehata, Amr S. Mahmoud, Kenneth Lundstrom, Debmalya Barh, Vasco Azevedo, Nagwa A. Sabri

**Affiliations:** 1Drug Research Centre, Cairo P.O. Box 11799, Egypt or mohamed.raslan@pharma.asu.edu.eg (M.A.R.); or sarah.raslan@pharma.asu.edu.eg (S.A.R.); eslam.mansour@drcbioeqs.com (E.M.S.); 2Department of Obstetrics and Gynecology, Faculty of Medicine, Ain Shams University, Cairo P.O. Box 11566, Egypt; amrsaadobygyn@med.asu.edu.eg; 3PanTherapeutics, CH1095 Lutry, Switzerland; lundstromkenneth@gmail.com; 4Department of Genetics, Ecology, and Evolution, Institute of Biological Sciences, Federal University of Minas Gerais (UFMG), Belo Horizonte 31270-901, Brazil; dr.barh@gmail.com (D.B.); vascoariston@gmail.com (V.A.); 5Institute of Integrative Omics and Applied Biotechnology (IIOAB), Nonakuri, Purba Medinipur 721172, West Bengal, India; 6Department of Clinical Pharmacy, Faculty of Pharmacy, Ain Shams University, Cairo P.O. Box 11566, Egypt

**Keywords:** nutrigenomics, antimicrobial resistance, novel antibiotics, gut microbiome

## Abstract

Nutrigenomics is the study of the impact of diets or nutrients on gene expression and phenotypes using high-throughput technologies such as transcriptomics, proteomics, metabolomics, etc. The bioactive components of diets and nutrients, as an environmental factor, transmit information through altered gene expression and hence the overall function and traits of the organism. Dietary components and nutrients not only serve as a source of energy but also, through their interactions with genes, regulate gut microbiome composition, the production of metabolites, various biological processes, and finally, health and disease. Antimicrobial resistance in pathogenic and probiotic microorganisms has emerged as a major public health concern due to the presence of antimicrobial resistance genes in various food products. Recent evidence suggests a correlation between the regulation of genes and two-component and other signaling systems that drive antibiotic resistance in response to diets and nutrients. Therefore, diets and nutrients may be alternatively used to overcome antibiotic resistance against novel antibiotics. However, little progress has been made in this direction. In this review, we discuss the possible implementations of nutrigenomics in antibiotic resistance against novel antibiotics.

## 1. Introduction

In contrast to nutrigenetics, which deals with the impacts on human health of the single-nucleotide polymorphisms (SNPs) that are associated with nutrient metabolism, nutrigenomics is concerned with studying the impacts of nutrients on gene expression to understand how foods influence human health using high-throughput technologies such as epigenomics, transcriptomics, etc. [1]. Nutrition plays an essential role in life since macro- and micronutrients are essential building blocks for sustaining life [2]. The pathways for nutrient–gene communication routes have existed throughout the evolution of life [3]. Furthermore, because some nutrients directly regulate the genome or the epigenome through interaction with transcription factors or chromatin modifiers, they have a health impact [4]. Thus, a nutrigenomics approach can provide a snapshot of genes that are turned on or off to regulate a trait or phenotype. Increasing research in this field should lead to a better understanding of how nutrition affects metabolic pathways and homeostasis regulation, which may then be used for the prevention of chronic diet-related disorders [5].

One of the first studies to use transcriptomics technology for human dietary interventions assessed the effects of a high-carbohydrate breakfast compared to a high-protein meal on the gene expression profile in blood leukocytes of healthy males [6]. Breakfasts high in carbohydrates resulted in the differential expression of genes mostly engaged in glycogen metabolism, whereas breakfasts high in protein content led to the differential expression of genes primarily involved in protein production [6]. The effects of the chronic feeding of various diets comprising vanaspati (rich in trans fatty acids (TFAs)), palm oil (rich in saturated fatty acids (SFAs)), and sunflower oil (rich in polyunsaturated fatty acids (PUFAs)) at a 10% level of all dietary composition on 11β-hydroxysteroid dehydrogenase type 1 enzyme (11β-HSD1) gene expression have been examined in rat retroperitoneal adipose tissue. When compared to a PUFA-enriched diet, meals high in TFAs and SFAs showed superior 11β-HSD1 gene expression in rat retroperitoneal white adipose tissue (RPWAT). The risk of developing obesity and insulin resistance increased because of the enhanced local conversion of inactive to active glucocorticoids in adipose tissue [7]. Recently, using a nutrigenomics approach, we showed that Indian dietary habits and food ingredients may have reduced the severity and death rate of the COVID-19 pandemic in Indians [8].

Colistin (polymyxin E) is a polymyxin antibiotic extensively used in animal health for the oral treatment of enterobacterial digestive infections in pigs, poultry, and cattle [9]. Previous studies utilizing culture-dependent techniques on farm animals in Tunisia revealed the prevalence of colistin-resistant Gram-negative bacteria. In a recent investigational study, DNA analysis for ten known *mcr* genes was conducted on cloacal swabs from 195 broiler chickens from six farms in Tunisia [10]. A total of 81 (41.5%) of the 195 animals tested positive for mcr-1 with positive cases on all farms. These findings corroborate the emergence of colistin resistance in farm animals in Tunisia and suggest that investigating antibiotic-resistance genes (ARGs) might contribute to epidemiological research on antimicrobial resistance dissemination [10].

A nutrigenomics study showed the positive effects of phytobiotics and organic acids on ghrelin gene expression levels, gut microbiota composition, performance metrics, and intestinal histomorphological alterations in broiler chicks compared to antibiotics [11]. The addition of phytobiotics enhanced (*p* < 0.05) villus height and the ratio of villus height/crypt depth in the ileum, jejunum, and duodenum, while decreasing ghrelin gene expression levels. Total coliform and *E. coli* levels in cecal and ileal digesta were considerably lower (*p* < 0.05) compared to antibiotic treatment. Lactobacillus spp. were shown to be favorably associated with the villus height/crypt depth ratio in the duodenum by correlation analysis. Based on the nutrigenomics methodology, the findings highlighted the significance of gene–nutrient–microbiota relationships. As a result, phytobiotics and organic acids may be viable alternatives to antibiotics for increased performance and immunity in chickens, as well as healthier meat production [11]. Moreover, phytobiotics will contribute to decreased antibiotic use and the prevention of the potential emergence of microorganisms resistant to antibiotics.

Metagenomics approaches have been employed to investigate the content and dynamic distribution of ARGs, as well as the microbial population, in three types of factory-processed Chinese garlic powder (GP) products [12]. The findings revealed that 126 ARG genes from 11 different ARG species were detected. With the processing of GP, the expression of ARGs increased at first and later decreased [12]. Since garlic is added to food, this finding raises concern about the possible emergence of antibiotic-resistant strains from dietary supplies.

This review aims at highlighting the effect of nutrients on the genome or epigenome which may lead to emerging antimicrobial resistance for novel antibacterial agents, which has been discussed using nutrigenomic approaches.

## 2. Methods

In the present review, the following sources were included: randomized controlled trials (RCTs), controlled non-randomized clinical trials (CCTs), retrospective and prospective comparative cohort studies, case-control or nested case-control studies, reviews, systematic reviews, and thematic books.

A search strategy was designed using medical subject headings (MeSH). The MeSH terms of supplements, nutrition, nutrigenomics, gut microbiome, and antimicrobial resistance were used to systemically search PubMed and MEDLINE databases. Only studies in the English language were included. All relevant publications up to April 2023 were included. No limits regarding study design or date were set for the search. Duplicate studies were removed from our study pool. All included studies were scanned against inclusion and exclusion criteria. Our inclusion criteria primarily focused on published literature that assessed the effect of nutrients on the genome and its correlation with emerging antimicrobial resistance for novel antibacterial agents.

## 3. Results and Discussion

### 3.1. The Effect of Nutrition on the Human Gut Microbiome

Trillions of bacteria occupy the human gut, forming a dynamic ecological system involved in both health and disease. The composition of the gut microbiota is unique to each individual and tends to be generally constant throughout life, but daily transitory changes are noted. Diet is a fundamentally modifiable element determining the makeup of the gut microbiota, implying that therapeutic dietary techniques are possible to control microbial diversity, composition, and stability. While food can cause changes in the gut microbiota, these changes appear to be transient. It is uncertain if long-term dietary changes may cause lasting changes to the gut microbiota, owing to a dearth of long-term human nutritional interventions or long-term follow-ups of short-term dietary treatments [13].

Aside from nutrition, the gut microbiota is influenced by a mix of extrinsic factors such as lifestyle and medicine and intrinsic factors such as host genetics, immunology, and metabolic control. It is commonly known that external variables show the greatest influence, with nutrition being the most extensively studied variable [14]. On the other hand, according to a 2016 study of 1126 twins, genetics shows a modest 8.8% average effect in determining the gut flora [15].

The human diet normally comprises many food components, such as protein, fat, and carbohydrates, with a proper balance required. The community and diversity of the gut microbiome have a large impact on the kind and quantity of nutrition. The digestion of food elements results in variation in the end products, which play an important role in the prevention, management, and treatment of diseases like cancer and diabetes [16,17]. Moreover, the flourishing or fading of the beneficial microbiome may result in increased metabolites in a healthy direction or the appearance of opportunistic genera. The production of certain metabolites may result in a pathogenic appearance that can influence host physiology and gene expression, resulting in the development of various diseases [18].

A fat-containing diet alters the gut microbial composition, usually resulting in a drop in Bacteroidetes and an increase in both Firmicutes and Proteobacteria [19]. Furthermore, certain genera of the class Gammaproteobacteria increased their abundance in comparison to other particular genera, resulting in a change in the microbial population and diversity [20]. These alterations reduce the microbial synthesis of short-chain fatty acids and antioxidants. Changes in metabolites have consequences, such as increased disease risk. Carbohydrates have been proven to boost cell survival in cancer by increasing the expression of genes related to fatty tissue and obesity [21]. Diets high in complex carbohydrates generated from plant tissue are slowly digested by the gut bacteria, particularly those found in the distal intestine. Plant fiber digestion enhances the symbiotic microbiota, which leads to an increase in short-chain fatty acids, which have a role in energy supply and hence in human health. Complex carbohydrates improve body weight, food intake, glucose homeostasis, and insulin sensitivity [22]. On the other hand, several studies have found a link between a higher-fiber diet and a lower risk of irritable bowel syndrome, inflammatory bowel disease, cardiovascular disease, diabetes, and colon cancer [23].

### 3.2. The Effect of Diets and Nutrients on Human-Genome-Level Expression

In the context of nutrigenetics and nutrigenomics, bioactive components are dietary ingredients that may transmit information from the external environment, alter gene expression in the cell, and hence affect the overall function of the organism (Figure 1). It is critical to consider nutrition not only as a source of energy and essential nutrients necessary for life and organism growth but also as a factor impacting biochemical processes, biochemical pathway activation, and health/disease. Dietary bioactive components influence gene expression through chromatin structural modifications, the activation of transcription factors via signaling cascades, or direct ligand binding to nuclear receptors [24]. Individual phenotypes can be altered by nutrient-induced gene expression. SNPs in a variety of genes involved in inflammation and lipid metabolism, on the other hand, can modify the bioactivity of essential metabolic pathways and mediators, as well as the capacity of nutrients to interact with these mediators and metabolic pathways [25].

Gene expression in human blood and other tissues has been demonstrated to be dependent on gender, age, and time of day. However, additional variables that may influence gene expression have not been well investigated. For example, it is unclear whether the fasting or feeding condition will provide a clear answer related to the association between gene expression in the blood and obesity [26]. Several recent studies have demonstrated the adaptability of biological networks by revealing rapid network rewiring in response to various environmental challenges. Fasting and feeding responses in rats have been studied with some success [27]. Other studies have demonstrated that circadian clock genes in mouse heart tissue are controlled differentially during food intake and fasting [28].


Figure 1Illustration of the relationships between nutrition and the genome [29]. (A) Nutrigenetics: genetic polymorphisms can induce differential gene expression. As a result, different metabotypes exist, which show different responses to nutrition, different nutrient requirements, and potential food intolerance. Of note, the location of SNPs can also affect epigenetic modifications. (B–D) Nutrigenomics: methyl donor availability, bioactivity of dietary compounds, and xenobiotics (B) can affect the one-carbon cycle and other pathways, thus consequentially, affecting DNA methylation and histone modifications (C). Not just parental molecules (B) but also derived compounds and metabolic products of microbial activity (D) can affect these pathways (C).
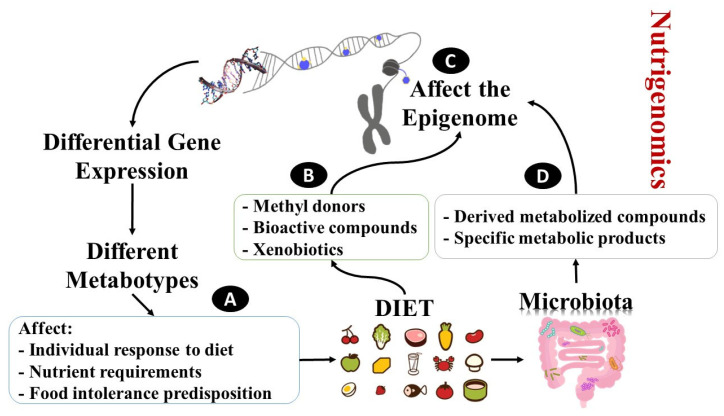



### 3.3. Antimicrobial Resistance Mechanisms

Antimicrobial resistance (AMR) is defined as the ability of bacteria to live and thrive in the presence of antimicrobial substances. Antimicrobial agents, such as antibiotics, disinfectants, and food preservatives, are available and can be employed against microorganisms to restrict their ability to grow, prevent their reproduction, or even kill them. Natural, semi-synthetic, and synthetic substances with distinct mechanisms capable of causing major changes on the metabolic and physiological levels of bacteria include cell-wall synthesis modifications such as β-lactams and glycopeptides, protein synthesis inhibitors such as macrolides and tetracyclines, metabolic pathway inhibitors such as sulfonamides, and interference with DNA replication and translation such as fluoroquinolones [30].

Antimicrobial resistance is a dramatic natural phenomenon that arises spontaneously over time through genetic change inside bacteria. However, variables like drug usage and misuse accelerate this transformation. For example, administering incorrect antibiotics for viral infections such as the flu might contribute to the resistance process. It poses a comparable hazard to animals and food production sustainability [30]. Antimicrobial resistance mechanisms are classified into four types: restriction of drug uptake, alteration of drug targets, inactivation of drugs, and active drug efflux (Figure 2). Intrinsic resistance mechanisms include drug target alterations, drug inactivation, and drug efflux; acquired resistance mechanisms include drug target modifications, drug inactivation, and drug efflux. Because of structural changes, etc., the processes utilized by Gram-negative bacteria differ from those used by Gram-positive bacteria [31].

### 3.4. Antimicrobial Effect on the Human Gut Microbiome

The gut microbiota is vital for host health and is influenced by a variety of variables, including antibiotics. Antibiotic-induced changes in gut microbial composition can have a negative impact on health by reducing microbial diversity, changing functional attributes of the microbiota, forming and selecting antibiotic-resistant strains, and increasing susceptibility to infection with pathogens like *Clostridioides difficile* [32].

Recently, scientists discovered that broad-spectrum antibiotics had a negative influence on the gut microbiome [33]. The gut microbiota, which is home to bacteria, archaea, microeukaryotes, and viruses, plays an important role in human health. It inhibits pathogen colonization, modulates gut immunity, supplies necessary nutrients and bioactive metabolites, and aids in energy balance [33]. It has been reported that oral antibiotics are associated with an increased risk of colon cancer [34]. For example, extremely preterm children subjected to continuous antibiotic therapy had fewer varied bacterial communities, lower species richness, and higher ARGs in their gut [35,36].

Antibiotics can cause antibiotic-associated diarrhea (AAD), and studies have shown that clindamycin might modify the microbial population, promoting the colonization of potential pathogens such as *C. difficile*, resulting in diarrhea and colitis [37]. The use of antibiotics has also been linked to changes in protein expression and energy consumption in the microbiota, with a minor increase following antibiotic therapy, possibly as a coping strategy for antibiotic stress, which decreases at later stages and after exposure to antibiotics [38].

Antibiotics can affect the transcription of numerous important functional genes, including those encoding transport proteins, glucose metabolism genes, and protein synthesis genes. Exposure of *Pseudomonas aeruginosa* to sub-inhibitory antibiotic doses elevated the expression of virulence-associated genes, resulting in the increased production of rhamnolipids and phenazines. Numerous investigations have indicated that aminoglycosides, lactams, vancomycin, and oxacillin may all promote biofilm development at sublethal doses [32]. These biofilms then serve as antibiotic-resistance reservoirs. It provides further resistance to bacteria against various antibiotics and host defense, making treatment difficult for people and causing a variety of complications such as pipe/equipment obstruction in healthcare settings and food sectors [39]. Metabolomics profiles were examined in antibiotic-treated pigs from postnatal days 7 to 42, subjected to a corn-soy baseline diet with or without infeed antibiotics [40]. The antibiotic-treated group exhibited increased amounts of metabolites linked to amino acid metabolism, resulting in lower amino acid concentrations. Short-chain fatty acid production was also reduced as butyrate and propionate levels declined [40].

### 3.5. Phylogenetic Groups and Antimicrobial Resistance Genes from Poultry

There is little information on the occurrence of metal and antibiotic resistance in potentially pathogenic *E. coli* entering the food chain from pigs, which might endanger human health. The phenotypic and genotypic resistance of *E. coli* to 18 antibiotics and three metals (mercury, silver, and copper) at pig slaughterhouses in the United Kingdom has been investigated, which revealed resistance to streptomycin, sulphonamide, oxytetracycline, ampicillin, chloramphenicol, ceftiofur, amoxicillin-clavulanic acid, trimethoprim-sulfamethoxazole, aztreonam, and nitrofurantoin [41]. *E. coli* isolated from meat originating from different animal species may have ARGs and therefore pose a risk to human health. To identify antimicrobial resistance genes in *E. coli* isolates from pig, cattle, chicken, and turkey meat and to determine if resistance genotypes are related to phylogenetic groupings or meat species, a normal culture procedure was applied, including 313 isolated *E. coli* samples [42].

Resistance genes could be detected by PCR in 98% of resistant isolates. The tetracycline resistance genes tet(A) and tet(B), streptomycin resistance genes strA and aadA1, sulphonamide resistance genes sulI and sulII, kanamycin resistance genes dfr and aphA, and ampicillin resistance genes blaTEM have been identified. One stx1-positive *E. coli* isolate recovered from pigs carried the tet(A) gene and belonged to the phylogenetic group B2, whereas another stx1-positive strain isolated from cattle was multi-resistant, tested positive for blaTEM, aphA, strA-B, sulII, and tet(A), and belonged to the phylogenetic group A. Most *E. coli* populations with diverse resistance genes to a single antibiotic showed statistically significant differences in MIC values [42].

The genomic backbone and plasmid correlations with antimicrobial resistance were investigated [43]. A total of 72 pathogenic avian *E. coli* (APEC) strains were studied. Isolates that were resistant to tetracycline and trimethoprim-sulfamethoxazole (87.5% each) and harbored blaTEM (61.1%) dominated. Furthermore, phylogroup D was the most common, in total at 23.6%, and was among multidrug-resistant (MDR) isolates (14/63). The results indicated that group D strains have a high capacity to host a wide range of plasmids (Inc-types) harboring various AMR genes. This means that phylogroup D might pose a problem in dealing with antimicrobial resistance in poultry [43].

### 3.6. New Antibiotics against Microbial Resistance

Antibiotic resistance and the growth of multidrug-resistant bacterial strains have now become extremely widespread in hospitals and pose a threat to worldwide infectious disease control. Possible antibiotic-resistance approaches are being examined, and different mechanisms for the antibiotic resistance of certain previously beneficial antibiotics are under investigation [44]. If an active substance does not exhibit cross-resistance to current antibiotics, it is considered novel. Cross-resistance is defined in this context as resistance to the same class of antibiotics that may be assessed by systematic, in vitro sensitivity testing of genetically determined bacteria. If sufficient information on cross-resistance is lacking or unavailable, an active substance is considered innovative if it belongs to a new class of antibiotics, has a novel target or binding site, or demonstrates a novel mechanism of action [44].

Since 2017, eight novel antibacterial active compounds have been authorized, including one for the treatment of tuberculosis (TB). The non-profit organization TB Alliance created pretomanid, a therapeutic compound for the treatment of multidrug-resistant TB. Approximately half of the new antibiotics licensed target carbapenem-resistant Enterobacteriaceae (CRE), oxacillinase-48-producing Enterobacteriaceae (OXA-48), and β-lactamase-producing Enterobacteriaceae (ESBL). Despite significant advances in TB and *C. difficile* research, antibiotics are still ineffective in treating carbapenem-resistant *Acinetobacter baumannii* and *Pseudomonas aeruginosa* [45].

Since 2019, the new chemical compound zoliflodacin has been evaluated in Phase III for the treatment of multidrug-resistant *Neisseria gonorrhoeae* [46]. It is the first antibiotic of the spiropyrimidinetrion class to be synthesized. Its novel action comprises inhibition of type II bacterial topoisomerase, targeting a different location than fluoroquinolones [47]. Zoliflodacin has a very low resistance frequency and is efficacious not only against multidrug-resistant *N. gonorrhoeae* with MICs ranging from 0.002 to 0.25 μg/mL but also against a variety of Gram-positive and Gram-negative bacteria [42]. Even when bacteria are subjected to combinations of zoliflodacin and antibiotics currently in use, such as ceftriaxone, doxycycline, and gentamycin, no resistant mutations of *N. gonorrhoeae* have been detected. Zoliflodacin does not exhibit cross-resistance with currently available fluoroquinolones [46].

Ridinilazole is a synthetic, fast-acting antibiotic from the bis-benzimidazole family for the treatment of *C. difficile* [48]. It is poorly absorbed by the gastrointestinal lumen after oral administration. Ridinilazole is virtually perfect for the treatment of CDI, as it showed in vitro selective efficacy against *C. difficile*, minimal systemic absorption, and a decreased impact on the gut microbiota. Ridinilazole-resistant strains have not been identified. The findings of the action on the microbiota are quite encouraging, as the microflora remained essentially intact following ridinilazole treatment. In contrast, the most recent fidaxomycin therapy showed altered gut microbiota, and vancomycin therapy resulted in a significant drop in *Bifidobacteria* [44].

Delafloxacin, a new antibacterial DNA and topoisomerase IV inhibitor fluoroquinolone, shows increased activity in acidic media [49]. Its comparable affinity to both DNA gyrase and topoisomerase IV in Gram-positive (*Staphylococcus aureus*) and Gram-negative (*E. coli*) bacteria limits the potential for antibiotic resistance, requiring the accumulation of numerous mutations in both enzyme genes [44]. Eravacycline is a completely synthetic fluorocycline of the tetracycline family that was engineered to treat complicated intra-abdominal infections (CIAI). It primarily combats the acquired resistance of regular tetracyclines [50]. The acquisition of genes encoding certain efflux pumps and the presence of ribosomal protection proteins (RPPs) are the two key mechanisms causing pathogen resistance to tetracyclines [51].

Various efflux pumps have been found in Gram-positive and Gram-negative bacteria. The most common efflux pumps are encoded by the tet(A) and tet(B) genes in Gram-negative bacteria and the tet(K) and tet(L) genes in Gram-positive bacteria [45]. Eravacycline has the same pharmacophore as tetracyclines, but it has two distinct modifications in ring D at positions C7 (addition of a fluorine atom) and C9 (addition of a pyrrolidine acetamide group) [52]. The modifications at positions C7 and C9 render eravacycline efficacious against Gram-positive and Gram-negative bacterial strains otherwise resistant to first- and second-generation tetracyclines. Eravacycline, like other tetracyclines, inhibits the entrance of molecules from the aminoacyl-tRNA complex by reversibly binding to the ribosomal 30S subunit. However, compared to typical tetracyclines, the interaction between eravacycline and ribosomes is significantly stronger due to the recognition of several target sites. The first- and second-generation compounds are bacteriostatic; however, eravacycline possesses in vitro bactericidal activity against selected strains of *A. baumannii*, *E. coli*, and *K. pneumoniae* [44].

Plazomicin is a novel aminoglycoside derived from a modified sisomicin (a particular antibiotic against Gram-negative infections for which gentamicin, the first-choice molecule, was ineffective) [53]. Plazomicin inhibits most of the aminoglycoside-modifying enzymes (AME) that inactivate aminoglycosidic drugs in Enterobacteria spp. due to its novel chemical structure in comparison to other aminoglycosides. It differs significantly from the structures of gentamycin and tobramycin but is similar to amikacin [44].

### 3.7. How Genomics Mitigates the Public Health Impact of Antimicrobial Resistance

Whole-genome sequencing (WGS) and, more recently, metagenomic investigations have considerably improved our understanding of the antimicrobial resistance (AMR) process, and these technologies are guiding mitigation measures for better understanding and controlling AMR (Table 1) [54]. Culture-based antimicrobial susceptibility testing (AST), which is still used in clinical microbiology and patient care, has historically been used to identify AMR. While phenotyping gives clear visual evidence of how bacteria will interact with an antibiotic, it typically provides little or no information about resistance mechanisms, with divergent genetic clones frequently exhibiting similar resistance profiles [55].

Multi-locus sequence typing (MLST), a genetic typing method, provides a better level of pathogen resolution than AST but is very limited since it only describes a tiny part of a genome. WGS, on the other hand, gives genome-wide information at the single nucleotide level that may be utilized to determine the existence and mechanisms of AMR, as well as pathogen identification, virulence, and origin [56,57]. With the advent of next-generation sequencing (NGS), which uses high-throughput, parallel sequencing of DNA fragments, pathogen genomes may now be identified quickly and at a cheap cost [58,59].

Comparative phylogenetic analysis can be used to determine the degree of relatedness between different isolates based on the extent of genome similarity and, when combined with epidemiological and clinical data, can help understand the specific temporal patterns of AMR and transmission [55]. Furthermore, recent advancements in metagenomic sequencing methods have completely eliminated the need for establishing bacterial cultures. As a result of combining all accessible genetic information in a sample, metagenomic analysis allows for a shift in focus from an individual pathogen to the community microbiome landscape, resulting in a highly comprehensive model of how pathogens interact, mobilize, and access AMR genes [60].


pharmaceuticals-16-01093-t001_Table 1Table 1Case studies on using whole-genome sequencing (WGS) to reduce the public health burden of antimicrobial resistance (AMR).Case 1: International Surveillance—Determination of the Population Structure and Epidemiology of Carbapenem-Resistant *K. pneumoniae* (CR-Kp) across Europe [61]JustificationWGS/WorkflowMain FindingsAdvantages of WGSThe primary reservoirs and transmission dynamics of CR-Kp in Europe are still poorly understood.For sequencing, European hospital laboratories have submitted consecutive clinical isolates of CR-Kp, along with a susceptible strain for comparison.Primary cause of CR-Kp dissemination (Carbapenemase acquisition);another main source of CR-Kp spread (nosocomial acquisition).A baseline for continuous CR-Kp monitoring.Emphasize the importance of nosocomial spread.**Case 2: Enhancing the National Surveillance of Antimicrobial Resistance in the Philippines** [55]
**Justification**

**WGS/Workflow**

**Main Findings**

**Advantages of WGS**
National laboratory-based surveillance showed an increase in AMR incidences over the preceding ten years. Understanding of the epidemiology and causes of AMR remained limited.Retrospective sequencing of MDR GNB collected before the introduction was performed and examined with phenotypic and epidemiological data.*E. coli* ST410, drivers of carbapenem resistance at several healthcare system levels were found, including a localized outbreak of plasmid-driven CR-Kp impacting a single healthcare facility.The implementation of efficient infection control methods was made.Improved global coverage.**Case3: Investigating an MRSA Outbreak in a Neonatal Unit in the UK** [62]
**Justification**

**WGS/Workflow**

**Main Findings**

**Advantages of WGS**
Over a 6-month period, phenotypically comparable MRSA isolates were found in patients in a special baby care unit but could not be connected chronologically or geographically.WGS was performed on all MRSA isolates received from special baby unit patients.MRSA isolates from the community, as well as screening samples from elsewhere in the hospital, were also sequenced.Two previously excluded isolates were part of the epidemic, allowing temporal linkages between patients to be established.Beyond the newborn unit, a large transmission network was discovered.Testing of a large number of isolatesPrecise identification of related strainsAllowing for comprehensive epidemic reconstruction.Allowing for the identification of the source of the epidemic and the successful implementation of infection control measures.**Case 4: Investigating the Direction of Transmission in an *A. baumannii* Outbreak in a UK Hospital** [63]
**Justification**

**WGS/Workflow**

**Main Findings**

**Advantages of WGS**
The molecular typing of a cluster of *A. baumannii* isolates acquired at a UK hospital suggested a clonal epidemic, but the route of transmission between cases could not be established.WGS analysis was performed on a group of isolates acquired from patients with similar molecular typing profiles and antibiograms.The index case was identified, and the subsequent chain of transmission was determined.One patient/isolate was found to be unconnected, and the outbreak investigation was abandoned.The directionality of transmission may be identified by WGS, allowing for a precise reconstruction of the outbreak.**Case 5: Contact Tracing and Detection of Secondary Cases of TB in the Netherlands** [64]
**Justification**

**WGS/Workflow**

**Main Findings**

**Advantages of WGS**
Secondary TB detection and screening are critical for TB control. The poor precision of molecular typing makes the accurate identification of case clusters and transmission networks difficult.Molecular typing and WGS.The two techniques were evaluated in terms of discrimination and accuracy.WGS proved more capable of determining the relatedness of isolates than molecular typing.Aided in the identification of transmission episodes.Contact tracing and generating a broader knowledge of TB control.**Case 6: Identifying the Drivers of AMR in Atypical Enteropathogenic *E. coli* (aEPEC) Strains Isolated from Children < 5 Years in Four Sub-Saharan African Countries and Three South Asian Countries** [65]
**Justification**

**WGS/Workflow**

**Main Findings**

**Advantages of WGS**
The incidence, causes, and drivers of AMR in *E. coli* intestinal isolates from children in the community in many places throughout the world were unclear.The phenotypic susceptibility of isolates and WGS were investigated and linked with antibiotic usage, disease state, phylogenetic lineage, and geographic location.AMR was shown to be prevalent, with 65% of isolates resistant to at least three antimicrobial medication classes.A wide spectrum of genetic pathways of AMR was discovered.Conduct a thorough examination of AMR across a vast geographical area.Revealing information about AMR epidemiology, distribution, and causes.**Case 7: Investigation of Colistin Resistance Detected in Commensal *E. coli* in Food Stock Animals in China** [66]
**Justification**

**WGS/Workflow**

**Main Findings**

**Advantages of WGS**
Routine surveillance revealed a significant increase in the rates of colistin resistance in bacteria colonizing pigs in China, but the cause of this resistance remained unknown.Conjugation tests were performed.The WGS on plasmids was utilized to identify the relevant gene.The plasmid-associated colistin resistance gene sequence was identified and named *mcr-1.*The genetic foundation of a novel AMR mechanism has been identified and characterized.**Case 8: Understanding of the Epidemiology of MDR and XDR Pathogens Amenable to Control by Vaccination** [67,68]
**Justification**

**WGS/Workflow**

**Main Findings**

**Advantages of WGS**
AMR is affecting the effectiveness of typhoid fever therapy. Resistance to azithromycin was discovered in Bangladesh and later in Pakistan, but the genetic basis and likelihood of spread remained unclear.WGS was used to examine clinical isolates of azithromycin-resistant S. Typhi. The phylogenetic analysis allowed the strains to be contextualized among contemporaneous S. Typhi isolates in both contexts.Resistant isolates in Bangladesh and Pakistan arose from the separate acquisition of mutations in the same gene.The breadth of azithromycin selection pressure and the critical need for disease management by vaccination.Two independent epidemics of azithromycin-resistant S. Typhi were identified.Development of innovative typhoid conjugate vaccines for infection control.


### 3.8. Potential Nutrigenomics Effects on Increased Antimicrobial Resistance against New Antibiotics

According to metagenomic research, commensal bacteria in healthy people help keep pathogenic bacteria at a low density, implying that carriage is not often a concern. When patients have invasive surgery, however, there is a loss of microbial diversity, which is followed by the colonization of harmful bacteria. The use of antimicrobials, which commonly leads to the selection of drug-resistant pathogens and enables horizontal gene transfer (HGT) of AMR genes between bacterial lineages and species, might increase this impact. WGS investigations are currently being utilized to uncover the colonization variables that allow specific infections to grow quickly and survive in such conditions, with the hope that treatments targeting persistent organisms will be developed to reduce pathogen colonization [69,70,71,72,73,74].

ARGs from heat-treated bacteria might be transmitted to other bacteria via a variety of HGT pathways (Table 2). HGT is aided by mobile genetic elements (MGEs) like plasmids, integrons, and transposons, which allow genes to travel more freely. The frequency of HGT is heavily influenced by the qualities of MGEs, the characteristics of the donor and recipient populations, and the environment. Three primary conventional HGT pathways have been indicated: (1) conjugation, (2) transformation, or (3) transduction. Other, less well-known processes of DNA transfer may occur [75].

Conjugation takes place between living bacterial cells and is not possible if the cells are heat-inactivated [76]. DNA fragments, including ARGs, may be released from lysed, heat-inactivated bacteria and transmitted through transformation. Natural transformation has been seen in over 60 bacterial species and is likely to occur in many more [77,78]. Few studies have specifically examined exogenous DNA absorption by bacteria in food. There is evidence that DNA stability is inversely proportional to DNA length and that while heat treatments destroy lysed-exposed ARGs, fragments may still be long enough to be transformed by other bacteria [79]. Transduction is a kind of HGT that is mediated by bacteriophages (phages) and related particles known as gene transfer agents (GTAs). Phages can package some genetic material (including ARGs) of their hosts by replicating within the host cell before lysing it (lytic) or by integration into the host cell genome (lysogenic) [76].


pharmaceuticals-16-01093-t002_Table 2Table 2Studies on the fate of antibiotic-resistance genes (ARGs) after exposure to heat.ProcedureMediumEvaluation Temperature (°C)SpeciesAntimicrobial Resistance Genes (ARGs) PresentStated Antimicrobial Resistance ProfilesRecipient SpeciesARGs DetectedPost-Treatment from Non-Culturable SamplesTransformation DemonstratedReferenceCooked—boiled (20 min), grilled (10 min), microwaved (5 min, 900 W), or autoclaved (20 min, 121 °C)Chicken, beef, porkNot Stated
*E. faecalis*

*aac(6′)-Ie-aph(2″)-Ia*
Aminoglycosides, except for streptomycin (predicted profile, not tested)
*E. faecalis*
YESNO[80]General heat treatmentsSaline40, 50,60, 70, 80, 90, 100
*E. coli*
*bla*CTX-M-1,*bla*CMY-2,*tet*A,*str*ACephalosporins, tetracycline, streptomycin
*E. coli*
YESYES70 °C for 30 min[81]Milk pasteurization (sterilization)Milk and elution buffer63.5, 121*S. aureus*,*S. sciuri**blaZ*,*mecC*,*tetK*Penicillin, methicillin, tetracycline
*S. aureus*
YESYES63.5 °C for 30 min[82]Non-food autoclavingDistilled water and in presence of salt121, 135Plasmid (pUC18)NSAmpicillin
*E. coli*
-YES121 °C for 15 min[83]


Individual patient risk factors for getting infected by extended-spectrum β-lactamase-producing bacteria include an extended hospital stay prior to infection, antibiotic exposure, and recent travel abroad (Figure 3) [68]. Male sex, older age, and co-morbidities are all risk factors for multi-drug-resistant (MDR) Gram-negative bacteria infections [69]. Changes in the human microbiota occur in response to disease, particularly when antibiotic exposure is frequent and/or chronic. Enterobacterales are common gastrointestinal colonizers that can serve as key reservoirs for mobile AMR genes [60].

Enterococci are innately resistant to cephalosporins, allowing them to develop unusually high densities in patients subjected to cephalosporin treatment and encouraging diffusion to other areas where infection occurs. The cognate response sensor (CroS) kinase and its cognate response regulator (CroR) are essential for cephalosporin resistance in *Enterococcus faecalis*, but little is known about the variables that govern this signaling system to modify resistance. To detect protein–protein interactions in *E. faecalis* cells, a protein fragment complementation assay was used, which revealed a previously unknown association of CroR with the HPr protein of the phosphotransferase system (PTS), which is responsible for carbohydrate uptake and catabolite control of gene expression. The potential of CroS to increase cephalosporin resistance and gene expression in a nutrient-dependent manner is limited by its connection with HPr, according to genetic and physiological investigations. Mutational studies revealed that the interface via which HPr associates with CroR differs from the interface via which it associates with other cellular partners [84].

A study in the United States showed that aminoglycosides were discovered to be the most common and widespread cause of AMR in healthy people, and aminoglycoside-O-phosphotransferases (aph3-dprime) were associated negatively with total calories and the consumption of soluble fibers. Data revealed that individuals with low ARGs ingested much more fiber in their diets than medium and high ARG individuals, who possessed a greater abundance of obligate anaerobes in their gut microbiota, particularly from the family Clostridiaceae. Moreover, machine learning was used to look for connections between 387 nutritional, physiological, and lifestyle characteristics and antimicrobial resistance and discovered that enhanced phylogenetic diversity of food was related to individuals with low ARG levels [85].

Data from animal studies showed that in both canines and felines, 23 (ARGs) were found in 50% of the samples, with tetracycline and aminoglycoside resistance genes being the most common. The abundance of a particular ARG tended to respond similarly to nutritional intervention. When compared to dogs subjected to a baseline diet, dogs on the high-protein and low-carbohydrate (HPLC) diet had a higher abundance of the tetracycline resistance genes tet(W), tet(O), and tet(44) and the macrolide resistance genes mefA and mel but a lower abundance of the β-lactam ARG CfxA6. The quantity of these ARGs was similar in HPLC-fed kittens and moderate-protein/moderate-carbohydrate (MPMC)-fed kittens [86]. The tetracycline resistance gene tet(W) was found in the greatest number of taxa, mostly in Firmicutes. Bifidobacterium, a genus widely utilized in dairy product fermentation and as a probiotic, shared tet(W) with a wide range of other species [86]. Dietary Cu and Zn supplementation in swine production may further increase the likelihood of antibiotic resistance spreading through the co-selection and mobilization of ARGs and subsequent transmission to humans [87,88]. Co-selection can occur when ARGs and metal-resistance genes (MRGs) are genetically linked (co-resistance), when the same resistance mechanism confers resistance to both metals and antibiotics (cross-resistance), or when a common regulator controls the expression of both metal- and antibiotic-resistance systems [89]. Indeed, increased dietary Cu and Zn dosages used to promote swine development have been demonstrated to select for Cu or Zn resistance and to co-select for antibiotic resistance in certain groups of swine gut bacteria containing harmful strains [90].

The previously mentioned new antibiotics in Section 3.6 showed no emerging resistance against their antibacterial activity, which makes them more adventitious than other existing antibiotics in different clinical settings. However, the aforementioned evidence regarding nutrigenomic effects on antimicrobial resistance suggests the potential emergence of resistance one day against these new antibiotics.

## 4. Conclusions

It is apparent that significant effort is being dedicated to the development of antibiotic alternatives to promote better clinical outcomes with a low incidence of resistance. As we move forward, we must keep resistance mechanisms in mind so that they can be perpetuated. Nutrigenomic pathways seem to carry a lot of unknowns that may result in the emergence of antibiotic resistance. The use of metagenomic analyses has greatly enhanced the understanding of the mechanisms underlying antimicrobial resistance (AMR), and the advanced technologies have facilitated the implementation of effective measures to better understand and manage AMR. Antibiotic-resistance genes (ARGs) from heat-treated bacteria might be transmitted to other bacteria via a variety of horizontal gene transfer pathways. For the moment, new antibiotics like zoliflodacin, ridinilazole, and eravacycline seem to overcome the occurrence of resistance, but one-day resistance against those agents may occur based on obtained nutrigenomic effect evidence.

## 5. Recommendations

Antimicrobial stewardship programs are encouraged and should be integrated with different nutrigenomic approaches in healthcare settings.Monitoring and limiting the use of new antibiotic molecules to overcome any potential incidence of antibiotic resistance.Focusing on the nutrient effect on human gut microbiome dysbiosis and its correlation with antibiotic-resistance incidence.

## 6. Limitations

This article did not cover the possible chances of integrating nutrigenomic approaches with clinical practice.There is no available data on emerging resistance against newly discovered antibiotic molecules.

## Figures and Tables

**Figure 2 pharmaceuticals-16-01093-f002:**
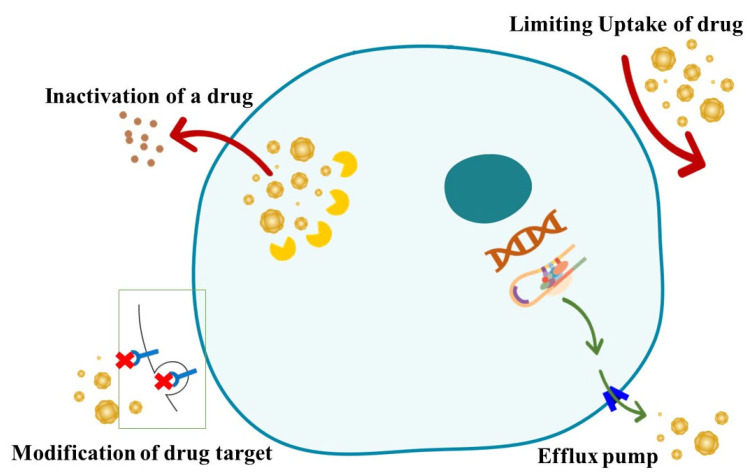
General antimicrobial resistance mechanisms [31]. By Kosmidis 2015. Reproduced from C Reygaert, W, 2018.

**Figure 3 pharmaceuticals-16-01093-f003:**
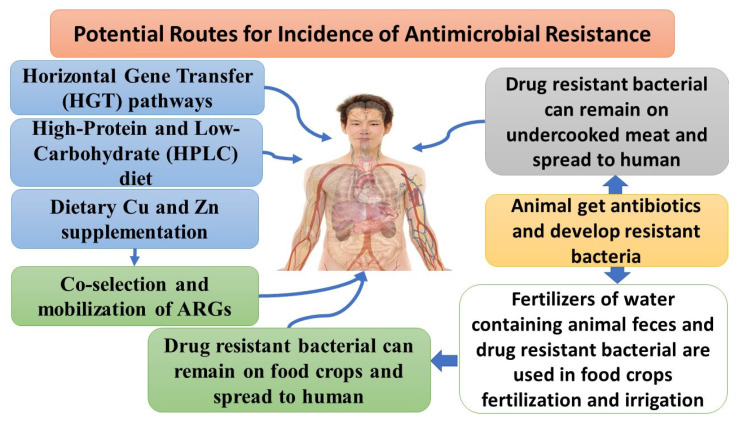
Potential routes for antimicrobial resistance.

## Data Availability

Data sharing not applicable.

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
