# Peer review of "Associations between Nutrigenomic Effects and Incidences of Microbial Resistance against Novel Antibiotics"

_pharmaceuticals, 2023, doi:10.3390/ph16081093_

Round 1
Reviewer 1 Report
The authors described the possibility that nutrigenomics effect of some molecules can influence the incidence of microbial resistant strains against novel antibiotics. The review give a new way to link two concepts, thus, is innovative. However, a certain improvement is necessary.
Major revisions:
1- The title put in relationship two not comparable worlds: the Nutrigenomics, a science, with the Incidence of Microbial Resistant Strains against Novel Antibiotics, a frequency. I propose to change the title by using the same words the authors use in the conclusion of abstract. Thus, the title, could become: "Association between Nutrigenomics approaches and Incidence of Microbial Resistant Strains against Novel Antibiotics". Or alternatively: "Association between some nutrigenomics effects and Incidence of Microbial Resistant Strains against Novel Antibiotics".
2- Line 41. The authors attempt a too generic and inadeguate definition of Nutrigenomics. In my opinion a good definition, genetically rigorous, is contained in the first paragraph of introduction of PMID: 33287559: please read, implement and cite.
3- Lines 64-65. The authors report some SNP and polymorphisms that are the manifest of genetic variability and are close to Nutrigenetics more than Nutrigenomic. It is important to clarify this difference by introducing a brief comment at the beginning of the paragraph. The paper PMID: 32545252 can clarify this difference and, if cited, can help the authors to solve this discrepancy.
4- The figures are well conceptualized but appear blurry. Please provide high quality figures.
Minor editing of English language is required
Author Response
Respected Reviewer 1
We wish to express our gratitude for your evaluation of our manuscript. Your valuable comments will certainly contribute to the improvement of the quality of our manuscript.
Comment #1:
The title put in relationship two not comparable worlds: the Nutrigenomics, a science, with
the Incidence of Microbial Resistant Strains against Novel Antibiotics, a frequency. I
propose to change the title by using the same words the authors use in the conclusion of
abstract. Thus, the title, could become: "Association between Nutrigenomics approaches
and Incidence of Microbial Resistant Strains against Novel Antibiotics". Or alternatively:
"Association between some nutrigenomics effects and Incidence of Microbial Resistant
Strains against Novel Antibiotics".
Reply:
We have changed the title to “Association between Nutrigenomics Approaches and
Incidence of Microbial Resistant Strains against Novel Antibiotics”.
-------------------------------------------------------------------------------
Comment #2:
Line 41. The authors attempt a too generic and inadequate definition of Nutrigenomics. In
my opinion a good definition, genetically rigorous, is contained in the first paragraph of
introduction of PMID: 33287559: please read, implement and cite
Reply:
We have used the PMID: 33287559 reference for the definition of nutrigenomics and the
new citation has been added (Reference 1).
-------------------------------------------------------------------------------
Comment #3:
Lines 64-65. The authors report some SNP and polymorphisms that are the manifest of
genetic variability and are close to Nutrigenetics more than Nutrigenomic. It is important
to clarify this difference by introducing a brief comment at the beginning of the paragraph.
The paper PMID: 32545252 can clarify this difference and, if cited, can help the authors to
solve this discrepancy.
Reply:
We totally agree with comment, and the PMID: 32545252 reference was used to clarify
the discrepancy between nutrigenetics and nutrigenomics, and the new citation has been
added (Reference 8).
-------------------------------------------------------------------------------
Comment #4:
The figures are well conceptualized but appear blurry. Please provide high quality figures.
Reply:
We have improved the resolution and quality of all figures.
-------------------------------------------------------------------------------
Comment #5:
Minor editing of English language is required
Reply:
The English language has been reviewed and edited.
-------------------------------------------------------------------------------

Reviewer 2 Report
1. Figure1, 2, 3 seem a little blurry. Resolution must be improved.
2. Consider creating a graphical abstract to increase article appeal.
3. Line 231 “Antimicrobial effect on human gut microbiome” consider revising as I feel you mean the effect of antimicrobial agents on…
Moderate editing of English language required
Author Response
Respected Reviewer 2
We wish to express our deep thanks for your effort in the evaluation of our manuscript. Your valuable comments suggestions were all performed and will certainly contribute to the improvement of the quality of our manuscript.
Comment #1:
Figures 1, 2, 3 seem a little blurry. Resolution must be improved.
Reply:
Kindly, Figures 1,2,3 are much better in resolution and fineness now in the updated version of the manuscript.
======================================================
Comment #2:
Consider creating a graphical abstract to increase article appeal.
Reply:
Kindly, a graphical abstract was done and uploaded on the system now.
======================================================
Comment #3:
Line 231 “Antimicrobial effect on human gut microbiome” consider revising as I feel you mean the effect of antimicrobial agents on…
Reply:
Kindly, as per recommended the sentence was rephrased in a correct meaning in the updated version of the manuscript.
=================================================

Reviewer 3 Report
Authors have meticulously compiled the available information on nutrigenomics and the incidence of microbial-resistant strains against novel antibiotics. The review is comprehensive and will serve as an excellent source of information on the topic. I only have a couple of minor observations:
1. There are some typos. The work would benefit from close editing.
2. Some paragraphs are too short (1 or 2 sentences), please fix them.
3. The aim of this work should be mentioned at the end of the introduction.
4. Figures 1 and 3 are not readable.
5. Table 1 is too wordy.
6. Rewrite the conclusions for clarity.
7. Rewrite the recommendation and limitations.
There are some typos. The work would benefit from close editing.
Author Response
Reviewer 3
We wish to express our deep thanks for your effort in the evaluation of our manuscript. Your valuable comments will certainly contribute to the improvement of the quality of our manuscript.
Comment #1:
There are some typos. The work would benefit from close editing.
Reply:
Kindly, typos and grammatical corrections are now corrected in the manuscript.
======================================================
Comment #2:
Some paragraphs are too short (1 or 2 sentences), please fix them.
Reply:
Kindly, short paragraphs in pages 4, 5, 7, 8, and 9 are now fixed.
======================================================
Comment #3:
The aim of this work should be mentioned at the end of the introduction.
Reply:
Kindly, the aim of this work is now mentioned at the end of the introduction (page 2).
======================================================
Comment #4:
Figures 1 and 3 are not readable.
Reply:
Kindly, figures 1 and 3 are now fixed and become clearer.
======================================================
Comment #5:
Table 1 is too wordy.
Reply:
Kindly, table 1 is now shortened in the updated version of the manuscript.
======================================================
Comment #6:
Rewrite the conclusions for clarity.
Reply:
The conclusions section has been revised and updated.
======================================================
Comment #7:
Rewrite the recommendation and limitations.
Reply:
Kindly, as per recommended, both the sections on Recommendations and Limitations were updated.
=================================================

Round 2
Reviewer 1 Report
The PMID 32545252 reference was proposed to clarify the discrepancy between nutrigenetics and nutrigenomics. The authors say that they have used this PMID but in the indicated rows I find the PMID of a their previous publication (marked as reference 8) which reports other arguments (COVID 19). It is not the recommended citation, nor any other citation that resolves the difference between nutrigenetics and nutrigenomics.
For all other suggestions, the authors improved the manuscript by accepting the reviewer's suggestions,
Author Response
Response to Reviewer-1
Comment #1
The PMID 32545252 reference was proposed to clarify the discrepancy between nutrigenetics and nutrigenomics.
Reply:
In the first review, the reviewer suggested to use PMID: 33287559 to define nutrigenetics and nutrigenomics. We used that reference in the revised version-1 as reference (1). However, now the reviewer asked to use another PMID 32545252 to clarify the discrepancy between nutrigenetics and nutrigenomics. We have used this PMID 32545252 now in this revised-2 version of the manuscript; here also the reference (1) is now PMID 32545252.
The sentence now reads as: first sentence under Introduction
“In contrast to nutrigenetics, which deals with the impacts on human health of the single nucleotide polymorphisms (SNPs) that are associated with nutrient metabolism;, nutrigenomics is concerned with studying impacts of nutrients on gene expression to understand how foods influence human health using high throughput technologies such as epigenomics, transcriptomics etc. [1].”
It is surprising that the suggested both the references by the reviewer is co-authored by Caradonna et al.
PMID: 33287559
Crit Rev Food Sci Nutr. 2022;62(8):2122-2139. Nutrigenetics, nutrigenomics and phenotypic outcomes of dietary low-dose alcohol consumption in the suppression and induction of cancer development: evidence from in vitro studies
Fabio Caradonna 1, Ilenia Cruciata 1, Claudio Luparello 1
PMID: 32545252
Nutrients. 2020 Jun 11;12(6):1748.
Science and Healthy Meals in the World: Nutritional Epigenomics and Nutrigenetics of the Mediterranean Diet
Fabio Caradonna 1, Ornella Consiglio 1, Claudio Luparello 1, Carla Gentile 1
In this review we are not trying to define the terms “nutrigenetics” and “nutrigenomics” which is already well known by the scientific community.
We also feel that there are better references to define and differentiate both the terms and don’t know what special in the suggested references by the reviewer.
There are lot of reference to define “nutrigenetics” and “nutrigenomics” some may be
https://pubmed.ncbi.nlm.nih.gov/21625170/
https://pubmed.ncbi.nlm.nih.gov/23113033/
https://link.springer.com/article/10.1007/s12291-017-0699-5
https://www.mdpi.com/1648-9144/55/6/283
https://www.sciencedirect.com/science/article/abs/pii/S006526602100016X
----------------------------------------------------------------------------------------
Comment #2
The authors say that they have used this PMID but in the indicated rows I find the PMID of a their previous publication (marked as reference 8) which reports other arguments (COVID 19). It is not the recommended citation, nor any other citation that resolves the difference between nutrigenetics and nutrigenomics.
Reply:
There is a misunderstanding. We have not used reference 8 to show the difference between nutrigenetics and nutrigenomics. This reference is cited in introduction to show how the nutrigenomics (transcriptome) has been used to understand how food ingredients may have regulated the COVID-19 severity and deaths.
Your recommended citation PMID: 32545252 which is now cited. Thanks.
----------------------------------------------------------------------------------------
Comment #3
For all other suggestions, the authors improved the manuscript by accepting the reviewer's suggestions,
Reply:
Thank you.
----------------------------------------------------------------------------------------

Reviewer 3 Report
The authors have made substantial improvements in the revised manuscript. I think it is ready for publication.
Author Response
Response to Reviewer-3
Comment
The authors have made substantial improvements in the revised manuscript. I think it is ready for publication.
Reply:
Thanks for your effort in reviewing our manuscript and your valuable comments which have improved the quality of the manuscript.
----------------------------------------------------------------------------------------
